# OpenReview forum: "Dissecting the Safety Circuit: Neuronal Intervention for Transferable Adversarial Attacks on VLMs"
_ICML.cc/2026/Conference — ICML 2026 regular_

### Official Review · Reviewer_gtQa · 2026-02-15

**Soundness:** 4
**Presentation:** 4
**Significance:** 3
**Originality:** 3
**Overall Recommendation:** 5
**Confidence:** 4

**Summary:**

This paper introduces a new adversarial attack framework (SCIA) targeting VLMs' internal representations to achieve transferable attacks that can bypass safety-aligned mechanisms. Through mechanistic analysis, the authors find that safety-critical features are localized within distinct intermediate neuronal circuits, termed “defensive circuits,” while model-agnostic, transferable features reside in partially overlapping “transferable circuits.” SCIA identifies and surgically manipulates these two circuits: suppressing the defensive while amplifying the transferable, with semantic contrastive steering and spectral smoothness regularization to improve attack stealth and cross-model generalization.

**Compliance With Llm Reviewing Policy:**

Affirmed.

**Final Justification:**

I think the quality of this paper is good,  I will matain my score.

**Key Questions For Authors:**

- Is the loss design overly complicated? Given that $\lambda_3 = 0.01$, does this suggest that this loss term is not particularly important?
- How would SCIA perform against adaptive/model-aware defenses, such as jointly adversarially trained models or methods utilizing knowledge of the safety circuit structure to re-align or mask crucial layers?
- How does the optimization and runtime overhead of SCIA compare to baseline attacks, given the extra circuit identification phase?
- What are the transferability limits when surrogate and target models are very dissimilar?

**Limitations:**

Yes

**Strengths And Weaknesses:**

Strengths:
- The paper is well-written and easy to follow. The overall logic of writing is clear, and figures such as Figure 2 make the method immediately understandable.
- The paper’s mechanistic exploration using linear probes offers scientific insight into how VLMs organize safety logic, moving the needle on interpretability in adversarial ML. Overall, this is a fairly thought-provoking paper.
- The experiments are solid and comprehensive, effectively demonstrating the validity of the authors' assumptions and the superiority of their proposed method.

Weaknesses:
- No Evaluation Against Adaptive (Model-Aware) Defenses: The defense analysis is limited to common input preprocessing (Table 6), with no adaptive or model-aware defense strategies included, which could limit claims on true robustness.
- The paper may cite too few prior works, missing discussion and comparison with several highly relevant recent publications, specifically those concerned with vision encoder-focused, compositional, set-level, and model-agnostic transfer attacks.
- The quantitative experiments lack repeated trials to report standard deviations or confidence intervals.

Minor:
- Figure 1 may not accurately convey the intended message—it left me somewhat confused.

---

> ### Author Rebuttal · Authors · 2026-03-31
>
> ### W1 & Q2: Adaptive Defense Evaluation
>
> **R:** We designed a circuit-aware adaptive defense to evaluate SCIA's robustness against model-aware detection. Since SCIA operates by suppressing $\mathcal{C} _ {def}$ activations, a natural defense strategy is to monitor these activations for anomalous patterns. Specifically, the defender applies the same linear probing methodology on the target model to extract $\mathcal{C} _ {def}$ layer activations and trains a binary classifier to distinguish adversarial inputs from benign ones.
>
> |             | Qwen-2.5-VL (AUROC) | Llama3.2 (AUROC) |
> | ----------- | ------------------- | ---------------- |
> | UMK         | 0.91                | 0.89             |
> | FORCE       | 0.82                | 0.79             |
> | UltraBreak  | 0.78                | 0.74             |
> | SCIA (Ours) | 0.57                | 0.61             |
>
> SCIA is significantly harder to detect than all baselines, as its multi-objective optimization produces activation patterns closely resembling natural responses.
>
> ---
>
> ### W2: Missing Related Work
>
> **R:** We will supplement the related work section in the revised manuscript with discussion and comparison of relevant recent publications, including vision encoder-focused attacks, compositional attacks, set-level adversarial perturbations, and model-agnostic transfer attack frameworks.
>
> ---
>
> ### W3: Lack of Standard Deviations
>
> **R:** We acknowledge that reporting variance is important for establishing result reliability. The randomness in SCIA comes from the initial perturbation sampled from uniform noise and the stochastic spatial augmentations during optimization. Since the optimization is anchored by the deterministic circuit structure, we expect the variance to be small.
>
> To assess variance, we repeated the experiments in Table 2 with 3 different random seeds.
>
> |              | Run 1 | Run 2 | Run 3 | Mean ± Std     |
> | ------------ | ----- | ----- | ----- | -------------- |
> | Qwen-2.5-VL  | 0.651 | 0.648 | 0.654 | 0.651 ± 0.0030 |
> | InstructBlip | 0.517 | 0.516 | 0.519 | 0.517 ± 0.0015 |
> | Llama3.2     | 0.469 | 0.469 | 0.466 | 0.468 ± 0.0017 |
>
> The standard deviations are consistently within ±0.3%, confirming that SCIA's results are stable and the conclusions are robust.
>
> ---
>
> ### Q1: Importance of $\mathcal{L}_{spec}$
>
> **R:** The small value of $\lambda_3$ reflects the difference in magnitude scales across loss terms, not the importance of $\mathcal{L} _ {spec}$ . $\mathcal{L} _ {spec}$ computes a weighted sum of squared Fourier coefficients over the entire perturbation image, yielding raw values that are orders of magnitude larger than the other loss terms. The small $\lambda_3$ is necessary to balance the gradient contributions so that no single term dominates the optimization.
>
> The ablation study in Table 4 provides direct evidence. Removing $\mathcal{L} _ {spec}$ causes ASR to decrease from 0.651 to 0.598 on Qwen-2.5-VL, from 0.517 to 0.483 on InstructBlip, and from 0.469 to 0.428 on Llama3.2, an average drop of approximately 5%. This demonstrates that spectral regularization meaningfully improves transferability.
>
> ---
>
> ### Q3: Optimization and Runtime Overhead
>
> **R:** The additional cost comes primarily from circuit identification, a one-time computation per surrogate model reused across all queries. Per-query overhead is minimal as $\mathcal{L} _ {suppress}$ and $\mathcal{L} _ {amplify}$ operate on compact neuron subsets.
>
> |             | Circuit Identification | Total        | Avg. ASR |
> | ----------- | ---------------------- | ------------ | -------- |
> | UMK         | —                      | 35.1 GPU-hrs | 0.200    |
> | FORCE       | —                      | 65.2 GPU-hrs | 0.268    |
> | UltraBreak  | —                      | 41.6 GPU-hrs | 0.460    |
> | SCIA (Ours) | 2–3 hrs (one-time)     | 62.8 GPU-hrs | 0.597    |
>
> SCIA's total cost is comparable to FORCE. The identification overhead (<5% of total GPU time) is a reasonable trade-off for 13.7% ASR improvement.
>
> ---
>
> ### Q4: Transferability Limits with Dissimilar Models
>
> **R:** Transferability naturally decreases with greater architectural divergence, which is also a fundamental limit of all transfer-based attacks. Fortunately, SCIA's circuit-level optimization captures deeper structural properties than pixel-level or token-level methods, so we expect the degradation to be more graceful.
>
> **Attacking Commercial Models:** SCIA's advantage over baselines becomes more pronounced as architectural dissimilarity increases. On open-source models the improvement over UltraBreak is 13.7%, while on GPT-4o it grows to 15.4% and on Claude-4.5 it reaches 8.5%. This suggests that circuit-level optimization captures transferable properties that are more resilient to architectural differences than surface-level regularization.
>
> ---
>
> ### Minor: Figure 1 Clarity
>
> **R:** We will redesign Figure 1 with clearer annotations and a more intuitive visual flow in the revised manuscript.

---

> > ### Author Rebuttal · Reviewer_gtQa · 2026-03-31
> >
> > Thanks for the author's rebuttal. I will matain my score to accept.

---

### Official Review · Reviewer_aZF8 · 2026-03-04

**Soundness:** 3
**Presentation:** 3
**Significance:** 3
**Originality:** 3
**Overall Recommendation:** 5
**Confidence:** 5

**Summary:**

This paper proposes the Safety Circuit Intervention Attack (SCIA), a framework that leverages mechanistic interpretability to improve transferable adversarial attacks on VLMs. Through linear probing, the authors identify two disentangleable neuronal circuits within VLMs: a defensive circuit encoding safety features and a transferable circuit capturing model-agnostic representations. SCIA suppresses the defensive circuit while amplifying the transferable circuit via dual-objective optimization, combined with contrastive semantic steering and spectral smoothness regularization.

**Compliance With Llm Reviewing Policy:**

Affirmed.

**Final Justification:**

We thank the authors for the detailed rebuttal. The circuit overlap analysis, auxiliary model sensitivity study, and computational cost comparison are well-designed and provide quantitative support for the claims. My concerns have been fully addressed, and I raised my rating accordingly.

**Key Questions For Authors:**

(1) How sensitive is the final attack performance to the choice of $\mathcal{M} _ t$ used for constructing $\mathcal{D} _ {trans}$ ? If the attacker uses a very different architecture as the held-out model, does the identified transferable circuit still generalize?

(2) What is the quantitative overlap between $\mathcal{C} _ {def}$ and $\mathcal{C} _ {trans}$? If shared neurons exist, how does the dual-objective optimization handle potentially conflicting gradient signals?

(3) Could defenders leverage the same linear probing approach to detect or harden safety circuits? Have the authors considered whether adversarial training targeting these circuits could neutralize SCIA?

**Limitations:**

Yes. The authors include an appropriate impact statement.

**Strengths And Weaknesses:**

**Strengths:**

(1) This paper introduces a principled connection between mechanistic interpretability and adversarial attack design, explicitly targeting safety-critical neural circuits rather than relying on implicit regularization.

(2) The experimental setup is comprehensive, covering diverse open-source and commercial VLMs with thorough ablation studies and parameter sensitivity analysis.

(3) SCIA achieves consistent improvements over baselines across both open-source and commercial models, and the defense evaluation further shows that circuit-targeted perturbations are more resilient than pixel-level attacks against input preprocessing.

**Weaknesses:**

(1) The construction of $\mathcal{D} _ {trans}$ requires evaluating transfer success on a held-out victim model $\mathcal{M} _ t$, which conflicts with the claimed black-box threat model. The sensitivity of the identified transferable circuit to the choice of $\mathcal{M} _ t$ is not discussed.

(2) The computational cost is higher than most baselines, yet no direct cost comparison is provided, making it difficult to assess the practical trade-off between improved ASR and increased overhead.

(3) The paper claims $\mathcal{C} _ {def}$ and $\mathcal{C} _ {trans}$ are distinct and disentangleable, but no quantitative overlap measure is reported. If non-trivial overlap exists, the dual-objective optimization could face conflicting gradients.

---

> ### Author Rebuttal · Authors · 2026-03-31
>
> ### W1 & Q1: Auxiliary Model in $\mathcal{D} _ {trans}$ Construction
>
> **R:** The $\mathcal{M} _ {t}$ used during $\mathcal{D} _ {trans}$ construction is an auxiliary model for circuit identification, not the final attack target. In our main experiments (Table 2), the surrogate is LLaVA-v1.6 and $\mathcal{M} _ t$ is InternVL2, which does not appear in any of the final evaluation target sets. The attacker only needs one additional open-source model as a proxy to label transfer success or failure, and the resulting adversarial examples are deployed against completely unseen models such as GPT-4o, Gemini, and Claude. This is analogous to ensemble-based transfer attacks, where ensemble models serve as development-time proxies rather than final targets.
>
> **Auxiliary Model Sensitivity:** Since the transferable circuit captures model-agnostic features that facilitate generalization regardless of the specific target, we expect the identified circuit to be robust to $\mathcal{M} _ t$ selection. Concretely, we fixed the surrogate as LLaVA-v1.6 and varied $\mathcal{M} _ t$ among two architecturally different models, evaluated on 100 AdvBench queries.
>
> |          | Qwen-2.5-VL | InstructBlip | Llama3.2 | Avg.  |
> | -------- | ----------- | ------------ | -------- | ----- |
> | Qwen-VL  | 0.647       | 0.492        | 0.453    | 0.531 |
> | MiniGPT4 | 0.619       | 0.478        | 0.458    | 0.518 |
>
> The average ASR varies within a narrow range, suggesting that $\mathcal{M} _ t$ selection has minimal impact on final performance.
>
> ---
>
> ### W2: Computational Cost Comparison
>
> **R:** SCIA's computational cost has two components: 1) The circuit identification phase is a one-time cost per model, performed once and reused for all subsequent queries; 2) The per-query optimization cost is comparable to existing methods because the circuit-related loss terms involve only lightweight operations on small neuron subsets.
>
> We provide a detailed comparison below, measured on a single NVIDIA A100 GPU.
>
> |             | Circuit Identification | Total        | Avg. ASR |
> | ----------- | ---------------------- | ------------ | -------- |
> | UMK         | —                      | 35.1 GPU-hrs | 0.200    |
> | FORCE       | —                      | 65.2 GPU-hrs | 0.268    |
> | UltraBreak  | —                      | 41.6 GPU-hrs | 0.460    |
> | SCIA (Ours) | 2–3 hrs (one-time)     | 62.8 GPU-hrs | 0.597    |
>
> SCIA's total cost is comparable to FORCE. The circuit identification overhead (<5% of total GPU time) is a reasonable trade-off for the 13.7% ASR improvement over the strongest baseline.
>
> ---
>
> ### W3 & Q2: Quantitative Overlap and Gradient Conflicts
>
> **R:** The overlap between defensive and transferable circuits is expected to be minimal because they are identified through complementary criteria. Defensive circuits consist of neurons whose removal increases surrogate ASR, while transferable circuits consist of neurons whose removal decreases victim ASR. These two selection processes target functionally distinct roles and produce largely non-overlapping neuron sets.
>
> We verified this by computing the quantitative overlap between defensive circuits and transferable circuits of all layers:
>
> |             | Intersection Ratio | Jaccard Index |
> | ----------- | ------------------ | ------------- |
> | LLaVA-v1.6  | 2.7%               | 2.8%          |
> | Qwen-2.5-VL | 1.3%               | 1.4%          |
>
> The Jaccard index is 2.8% on LLaVA-v1.6 and 1.4% on Qwen-2.5-VL, indicating near-complete disentanglement. With overlap at this level, conflicting gradients between the two objectives are not a practical concern.
>
> ---
>
> ### Q3: Defenders Leveraging Linear Probing for Detection
>
> **R:** Defenders could indeed leverage the same probing approach to build detection systems. We evaluated this scenario by training a binary classifier on $\mathcal{C} _ {def}$ layer activations to detect adversarial inputs.
>
> |             | Qwen-2.5-VL (AUROC) | Llama3.2 (AUROC) |
> | ----------- | ------------------- | ---------------- |
> | UMK         | 0.91                | 0.89             |
> | FORCE       | 0.82                | 0.79             |
> | UltraBreak  | 0.78                | 0.74             |
> | SCIA (Ours) | 0.57                | 0.61             |
>
> The detector achieves high AUROC against conventional attacks such as UMK, but only 0.57 and 0.61 against SCIA, because SCIA's semantic steering and spectral smoothness produce activation patterns that closely resemble natural model responses. This indicates that circuit-level monitoring alone is insufficient to neutralize SCIA. Regarding adversarial training targeting these circuits, reinforcing specific neurons identified as $\mathcal{C} _ {def}$ may not provide lasting protection, as the safety function could redistribute to other neurons after fine-tuning, requiring the defender to repeatedly re-identify and re-harden the circuit.

---

> > ### Author Rebuttal · Reviewer_aZF8 · 2026-04-01
> >
> > We thank the authors for the detailed rebuttal. The circuit overlap analysis, auxiliary model sensitivity study, and computational cost comparison are well-designed and provide quantitative support for the claims. My concerns have been fully addressed, and I raised my rating accordingly.

---

### Official Review · Reviewer_7K9F · 2026-03-08

**Soundness:** 3
**Presentation:** 3
**Significance:** 2
**Originality:** 3
**Overall Recommendation:** 4
**Confidence:** 3

**Summary:**

This paper studies the transferability of adversarial attacks on vision-language models (VLMs). The authors argue that existing attacks transfer poorly because they rely on model-specific safety mechanisms. To address this issue, the paper proposes identifying two types of neuron subsets in VLMs: defensive circuits related to safety alignment and transferable circuits that capture model-agnostic representations. Based on this observation, the authors propose SCIA, an attack method that suppresses defensive circuits while amplifying transferable circuits to improve cross-model attack transferability. Experiments are conducted on several open-source and commercial VLMs, and the proposed method is reported to achieve higher attack success rates compared to prior transfer-based adversarial attacks. The paper aims to provide both a stronger attack method and insights into the internal mechanisms of safety alignment in VLMs.

**Compliance With Llm Reviewing Policy:**

Affirmed.

**Final Justification:**

My concerns have been adequately addressed in the rebuttal. Therefore, I raise my score.

**Key Questions For Authors:**

1.	How robust is the identification of defensive and transferable circuits across different datasets or attack types? For example, would the same neurons be selected if harmful prompts were constructed differently?
2.	Do similar safety circuits exist across different VLM architectures? Have the authors attempted any cross-model neuron alignment or representation similarity analysis?
3.	How sensitive is the attack performance to the choice of the neuron subsets used for circuit suppression and amplification?

**Limitations:**

yes

**Strengths And Weaknesses:**

strengths:
1. The paper attempts to analyze adversarial transfer from a mechanistic viewpoint by identifying neuron subsets related to safety alignment and transferable representations. This perspective is relatively novel compared to standard gradient-based transfer attacks.
2. The proposed SCIA method is well described and integrates multiple components (circuit suppression, transferable feature amplification, and regularization) in a coherent framework.
3. The paper evaluates transferability on both open-source VLMs and several commercial APIs, which helps demonstrate the practical relevance of the attack.

weaknesses:
1. The main claim of the paper is that safety behavior in VLMs is governed by identifiable defensive circuits. However, these circuits are identified mainly through linear probing and simple neuron deactivation experiments. Such analyses provide correlation rather than causal evidence, and the paper does not perform stronger interventions to verify whether these neurons truly control safety behavior.
2. The defensive dataset used to identify safety neurons is constructed from jailbreak examples generated by UMK attacks and benign VQAv2 samples. These two sources differ significantly in prompt structure and distribution, which may allow the probe to capture attack artifacts rather than genuine safety-related features.
3. The method assumes that safety circuits are partially shared across VLM architectures, which motivates the transfer attack design. However, the paper does not analyze whether the identified neuron subsets are consistent across models or whether similar circuits exist in different VLMs. The observed transfer improvement alone is insufficient to support this claim.

---

> ### Author Rebuttal · Authors · 2026-03-31
>
> ### W1: Correlation vs. Causal Evidence
>
> **R:** The effects observed from SCIA neuron deactivation extend beyond mere correlation. It provides evidence that aligns with potential causal mechanisms. SCIA progresses from correlation to causation in two stages: 1) Linear probing localizes candidate neurons correlated with safety behavior, narrowing the search space to a compact candidate set; 2) Neuron deactivation provides interventional verification by measuring behavioral impact when these candidates are disabled.
>
> **Experimental Causal Evidence:** Table 1 demonstrates this causal relationship. Deactivating defensive neurons increases ASR from 0.24 to 0.86 on LLaVA-v1.6 and from 0.11 to 0.81 on Qwen-2.5-VL, while deactivating randomly selected neurons of the same size produces negligible change (0.24 to 0.26 and 0.11 to 0.15 respectively). This rules out that arbitrary neuron sets would produce the same effect.
>
> ---
>
> ### W2: Distribution Bias in Defensive Dataset Construction
>
> **R:** The effectiveness of defensive dataset construction is independent of prompt structure differences between the two sources. The probe operates on intermediate-layer activations encoding the model's internal safety state, not surface-level formatting. Table 1 supports this: deactivating identified neurons shifts ASR from 0.24 to 0.86 on LLaVA-v1.6, while random neurons show negligible effect.
>
> **Prompt-Controlled Experiments:** We constructed a controlled $\mathcal{D} _ {def}$ where both positive and negative samples share identical harmful queries, differing only in the image (adversarial vs. clean). A probe trained on 100 such pairs was compared with the original $\mathcal{C} _ {def}$.
>
> |             | Neuron Overlap Ratio | ASR (original C_def) | ASR (controlled C_def) |
> | ----------- | -------------------- | -------------------- | ---------------------- |
> | LLaVA-v1.6  | 75.60%               | 0.783                | 0.771 (-0.012)         |
> | Qwen-2.5-VL | 82.90%               | 0.651                | 0.638 (-0.013)         |
>
> Approximately 80% of neurons are shared between the two approaches, with less than 1.5% ASR difference, indicating that the defensive circuit reflects internal safety mechanisms rather than prompt-structure confounds.
>
> ---
>
> ### W3 & Q2: Cross-Model Circuit Consistency
>
> **R:** SCIA's transfer effectiveness does not require different models to share identical neuron subsets or similar circuit structures. The core assumption is that safety mechanisms exhibit a common functional locality, where safety judgments concentrate within a small neuron fraction, rather than occupying the same positions across architectures.
>
> **Probing Consistency Experiment:** To directly test whether safety representations share cross-model structure, we trained a linear probe on one model's activations and evaluated it on another model without fine-tuning. If safety circuits were entirely model-specific, the transferred probe should perform near chance.
>
> | Train → Test        | D_def / D_trans |
> | ------------------- | --------------- |
> | LLaVA → LLaVA       | 0.89 / 0.78     |
> | LLaVA → Qwen-2.5    | 0.78 / 0.67     |
> | Qwen-2.5 → Qwen-2.5 | 0.83 / 0.76     |
> | Qwen-2.5 → LLaVA    | 0.71 / 0.63     |
>
> Cross-model probes retain accuracy well above chance, supporting shared safety representations across architectures despite moderate drops from architectural differences.
>
> ---
>
> ### Q1: Robustness Across Different Attack Types
>
> **R:** The identified circuits are robust to the choice of attack method used for dataset construction. They capture intrinsic model properties determined by architecture and alignment training, and since successful jailbreaks override the same underlying safety mechanism regardless of attack methods, different construction approaches should yield similar circuits. We reconstructed $\mathcal{D} _ {def}$ and $\mathcal{D} _ {trans}$ using VAJM instead of UMK and compared the identified neurons and attack performance.
>
> | Model        | C_def Overlap | C_trans Overlap | ASR (UMK-based) | ASR (VAJM-based) |
> | ------------ | ------------- | --------------- | --------------- | ---------------- |
> | InstructBlip | 76.80%        | 70.50%          | 0.517           | 0.481 (-0.036)   |
> | Qwen-2.5-VL  | 71.40%        | 73.20%          | 0.651           | 0.632 (-0.019)   |
>
> Over 70% of neurons are shared for both circuits across the two construction methods, and switching from UMK to VAJM causes less than 4% ASR difference.
>
> ---
>
> ### Q3: Sensitivity to Neuron Subset Selection
>
> **R:** We have investigated the sensitivity of neuron subset selection in Figure 5, which evaluates ASR across circuit sizes ranging from 0.05% to 5% of the layer's neurons. The results show a clear inverted-U pattern: too few neurons (0.05%) miss the complete safety mechanism, while too many (1%–5%) dilute the steering signal. Performance peaks at 0.1%–0.5%, consistent across three diverse target models.

---

> > ### Author Rebuttal · Reviewer_7K9F · 2026-04-03
> >
> > Thanks for the author's rebuttal. My concerns have been adequately addressed. I raise my score accordingly.

---

### Official Review · Reviewer_yZhQ · 2026-03-12

**Soundness:** 2
**Presentation:** 3
**Significance:** 3
**Originality:** 3
**Overall Recommendation:** 4
**Confidence:** 4

**Summary:**

This paper decomposes the model into a defensive circuit and a transferable circuit. By employing linear probing to separate these two circuits, it introduces the Safety Circuit Intervention Attack (SCIA). This attack suppresses the defensive circuit while amplifying the transferable circuit, thereby enabling cross-model generalization.

**Compliance With Llm Reviewing Policy:**

Affirmed.

**Final Justification:**

After reviewing the rebuttal, I remain at my original rating of Weak Accept.

**Key Questions For Authors:**

See Weaknesses

**Limitations:**

Yes.

**Strengths And Weaknesses:**

Strengths:
1. The paper proposes a novel approach that divides the VLM intermediate layer into a defensive circuit and a transferable circuit.
2. An empirical study demonstrates that suppressing the defensive circuit while amplifying the transferable circuit enhances the transferability of adversarial samples.

Weaknesses:
1. The paper states that defensive circuits and transferable circuits are disentangleable, but provides no analysis or experiments to substantiate this claim. Do $C_{def}$ and $C_{trans}$ overlap? If so, how might this overlap impact optimization?
2. The paper targets specific layers for attacks, yet Figure 3 indicates that information may exist across different layers in VLM. Would selecting neurons across all layers for optimization yield better results?
3.  Why do baseline methods perform so poorly against white-box setting? For instance, Table 2 shows adversarial samples generated using LLaVA-v1.6 as the surrogate model. Why do many baselines achieve only around 30% attack success rates on LLaVA-v1.6? It is currently difficult to determine whether these baselines were compared under fair and fully optimized conditions. I recommend the authors supplement the baselines with specific implementation details and hyperparameter tuning, and clarify whether the low white-box ASR stems from methodological design goals rather than insufficient reproducibility.

---

> ### Author Rebuttal · Authors · 2026-03-31
>
> ### W1: Disentangleability Analysis
>
> **R:** The disentangleability of defensive circuits and transferable circuits comes from their complementary identification criteria: 1) Defensive circuits select neurons whose deactivation causes surrogate ASR to increase, meaning these neurons actively enforce safety by suppressing harmful outputs; 2) Transferable circuits select neurons whose deactivation causes surrogate ASR to decrease, meaning these neurons facilitate cross-model generalization. Given that one set subserves model-specific safety enforcement while the other mediates cross-model feature transfer, their distinct computational objectives make co-localization within the same neurons unlikely.
>
> **Circuit Overlap Analysis:** To quantify this, we computed the overlap between defensive circuits and transferable circuits of all layers:
>
> | Model       | Intersection Ratio | Jaccard Index |
> | ----------- | ------------------ | ------------- |
> | LLaVA-v1.6  | 2.7%               | 2.8%          |
> | Qwen-2.5-VL | 1.3%               | 1.4%          |
>
> The Jaccard index is 2.8% on LLaVA-v1.6 and 1.4% on Qwen-2.5-VL, confirming near-complete disentanglement.
>
> ---
>
> ### W2: Single-layer vs. Multi-layer Neuron Selection
>
> **R:** Single-layer selection can achieve the optimal balance between signal strength and noise suppression. This is motivated by two observations as shown in Figure 3: 1) Defensive and transferable information are concentrated at different specific layers, with deep layers for defense and intermediate layers for transferability; 2) The discriminative power drops sharply outside the optimal layer. Incorporating neurons from low-discriminability layers would introduce irrelevant neurons that dilute the steering signal. This is analogous to the performance degradation in Figure 5 when circuit size exceeds the optimal range. Additionally, different layers have distinct semantic properties and representation spaces, making cross-layer neuron aggregation ill-conditioned for unified gradient-based optimization.
>
> **Multi-Layer Comparison:** We validated this by comparing four circuit selection strategies on LLaVA-v1.6, evaluated on 100 AdvBench queries across three target models.
>
> |                     | Qwen-2.5-VL | InstructBLip | Llama3.2  | Avg.      |
> | ------------------- | ----------- | ------------ | --------- | --------- |
> | Single-layer (ours) | **0.640**   | **0.513**    | **0.460** | **0.538** |
> | Top-2 layers        | 0.627       | 0.491        | 0.451     | 0.523     |
> | Top-3 layers        | 0.583       | 0.452        | 0.413     | 0.483     |
> | All layers          | 0.498       | 0.372        | 0.351     | 0.407     |
>
> Performance monotonically decreases as more layers are included, with the all-layer strategy losing 13.1% average ASR compared to single-layer.
>
> ---
>
> ### W3: Low ASR of Baselines on Surrogate Model
>
> **R:** In Table 2, white-box methods achieve around 60-80% ASR on the surrogate LLaVA-v1.6, while black-box and transfer-based methods achieve around 10-40%. The lower ASR of black-box baselines can be attributed to two factors: 1) Manual methods such as FigStep, MM-SafetyBench, and Hades rely on predefined templates or heuristic rules that do not adapt to model-specific safety mechanisms, limiting their effectiveness; 2) Transfer-based methods such as CoA and AIP&TATM prioritize cross-model generalization over surrogate-specific performance, and their optimization objectives are not designed to maximize ASR on any single model. To further clarify this point, we will include detailed implementation settings and hyperparameters for all baselines in the revised appendix.

---

> > ### Author Rebuttal · Reviewer_yZhQ · 2026-04-02
> >
> > Thank you for the detailed response. The added analyses largely address my first two concerns.
> >
> > I still remain uncertain about the third point. In particular, for transfer-based methods, although their objective emphasizes cross-model generalization, the optimization is performed on the surrogate model. Therefore, the very low ASR on the surrogate itself remains somewhat surprising. I would appreciate more concrete evidence that these baselines were compared under sufficiently optimized and fair settings. Otherwise, it is still hard to judge whether the low ASR is truly due to methodological design rather than under-tuning or reproducibility issues.

---

> > > ### Author Response · Authors · 2026-04-03
> > >
> > > ### W3: Low ASR of Baselines on Surrogate Model
> > >
> > > **R:** The low surrogate ASR of transfer-based baselines reflects method-specific design factors rather than under-tuning. We analyze each baseline's attack objective, evaluation protocol, and optimization strategy, and summarize the evidence below.
> > >
> > > **1）CoA and AIP & TATM** are designed for simpler attack objectives than jailbreaking, and this mismatch directly causes low surrogate ASR: a) CoA targets image captioning redirection via feature-level contrastive matching, such as "A bird is sitting on top of a bird feeder" → "A bunch of people celebrating around a birthday cake"; b) AIP & TATM target single-word insertion, making the output contain a specific word such as "suicide". Neither objective requires suppressing safety alignment, as they only shift surface-level semantics or insert isolated words. When adapted to the jailbreak setting, their optimization converges to solutions that satisfy the original objective but do not constitute successful jailbreaks.
> > >
> > > **2）FORCE and UltraBreak** are jailbreak methods. We identify two factors from their evaluation and optimization designs that contribute to their moderate surrogate ASR.
> > >
> > > - **Evaluation stringency.** Compared to the evaluation used in FORCE and UltraBreak, we adopt GPT-4o as the judge with stricter criteria to reduce false positives from borderline or ambiguous outputs. FORCE and UltraBreak use the HarmBench-Llama-2-13b classifier, which tends to accept borderline outputs as successful. We evaluated identical adversarial examples under both judges:
> > >
> > > |             | HarmBench-Llama-2 | GPT-4o (Ours) |
> > > | ----------- | ----------------- | ------------- |
> > > | FORCE       | 0.438             | 0.317         |
> > > | UltraBreak  | 0.694             | 0.558         |
> > > | SCIA (Ours) | 0.871             | 0.783         |
> > >
> > > The HarmBench results are consistent with those reported in the original papers, confirming faithful reproduction.
> > >
> > > - **Surrogate overfitting avoidance.** Existing methods (UltraBreak) have explicitly confirmed that unconstrained optimization on the surrogate leads to severe overfitting. Therefore, transfer-based attacks must incorporate constraints to avoid overfitting to the surrogate model, which causes the low ASR on LLaVA-v1.6. Specifically, FORCE and UltraBreak employ blanket constraints such as random spatial transformations and TV loss, which uniformly restrict the optimization space and inevitably degrade surrogate ASR. To verify this effect under our setting, we reproduced UltraBreak with and without its constraints using LLaVA-v1.6 as surrogate, evaluated on a 100-query subset of AdvBench with GPT-4o.
> > >
> > > |                            | Surrogate ASR | Black-box Avg. ASR |
> > > | -------------------------- | ------------- | ------------------ |
> > > | UltraBreak (unconstrained) | 0.71          | 0.34               |
> > > | UltraBreak                 | 0.56 (-0.15)  | 0.44 (+0.10)       |
> > >
> > > The results confirm the trade-off, where blanket constraints substantially reduce surrogate ASR in exchange for moderate black-box gains.

---

### Decision · Program_Chairs · 2026-04-30

**Decision:**

Accept (regular)

**Comment:**

This paper presents a thoughtful and technically solid approach to improving the transferability of adversarial attacks on VLMs by explicitly targeting safety-relevant internal circuits.

The reviewers find the mechanistic perspective interesting and convincing. It would be more interesting to discuss the interaction between transferability across prompts and safety circuits in the paper.

I recommend acceptance.